# Multi-Network Asynchronous TDOA Algorithm Test in a Simulated Maritime Scenario

**DOI:** 10.3390/s20071842

**Published:** 2020-03-26

**Authors:** Ciro Gioia, Francesco Sermi, Dario Tarchi

**Affiliations:** Joint Research Centre (JRC), European Commission, Via E. Fermi 2749, I-21027 Ispra, Italy; francesco.sermi@ec.europa.eu (F.S.); dario.tarchi@ec.europa.eu (D.T.)

**Keywords:** TDOA, asynchronous nodes, AIS verification

## Abstract

In the last few years, the number of applications relying on position of vessels at sea has grown significantly. Usually, these applications exploit information provided by the Automatic Identification System (AIS). Unfortunately, the cooperative nature of AIS makes it vulnerable to different types of attack. Therefore, especially for critical applications, the veracity of the position information reported in the AIS message needs to be verified. Several techniques can be adopted to this end. This paper presents a mathematical extension of the traditional Time Difference Of Arrival (TDOA) localisation technique allowing merging TDOA measurement from synchronous and non-synchronous receivers. This technique was tested in a simulated scenario, where the position of a moving target was estimated using different configurations of the receivers network. The robustness of the proposed algorithm with respect to the traditional one is demonstrated. The proposed approach, which is derived form satellite applications, is not limited to the AIS signals or to the maritime domain, and it can be adopted to estimate the position of any radiofrequency transmitter, by employing a suitable number of non-synchronous receivers.

## 1. Introduction

Chapter five of the Convention for Safety of Life at Sea (SOLAS) [1] defines navigational systems and equipment necessary to guarantee the minimum safety standards in the operation of merchant ships. In 2000, the International Maritime Organization (IMO) adopted an additional requirement, imposing to all ship to carry an AIS device automatically broadcasting information about the ship. Specifically, the information includes identity, position, course, speed and other safety-related data. Besides transmitting its own data, the Automatic Identification System (AIS) system can receive information from other ships. Ships shall maintain AIS in operation at all times, except where international agreements are in place.

Originally designed for collision avoidance, the AIS allows global tracking of vessels, thanks to the recent expansion of terrestrial networks and satellite constellations of receivers. The growth of the applications relying on AIS highlighted the need to verify data-trustworthiness of its information. In fact, the cooperative nature of the system and the lack of intrinsic security make it vulnerable to false or missing declarations. Moreover, the AIS relies on sub-systems that are exposed to various types of attacks. In particular, AIS exploits positioning based on Global Navigation Satellite System (GNSS), which can be disrupted by both natural phenomena, e.g., ionosphere disturbances, and man-made interferences such as spoofing and jamming [2].

Several solutions have been proposed to prevent the spoofing of the connected GNSS [3]. Nevertheless, further vulnerabilities exist in the AIS communications, since AIS messages are sent in an unencrypted and unsigned form, making them trivial to intercept and modify. A possible solution to this issue has been proposed in [4]. Moreover, the AIS protocol does not require any authentication, hence a malicious user could easily impersonate any other vessel and broadcast AIS messages that will be treated as genuine by all receiving vessels. In [5], the authors proposed a secure ship-to-ship information sharing scheme to provide reliable communication between ships and between ships and Vessel Traffic Service (VTS). However, most approaches to enhance AIS trustworthiness focus on increasing the protection of the on-board hardware from intentional tampering or on the improvement of AIS protocol and scheme.

In this paper, a technique to check the veracity of AIS vessel positioning is presented. This technique allows the comparison of the position reported in the AIS message with that estimated performing a trilateration of the received AIS signals. No additional hardware or upgrades of the AIS protocol are required. The signal trilateration is based on the Time Difference of Arrival (TDOA) measurements, which have been extensively documented for radiolocation applications. However, the TDOA approach presents limitations mainly due to the limited coverage given by small networks of receiving stations and/or poor timing synchronisation among nodes.

To fill these gaps, an improvement of the traditional TDOA approach, namely ‘Multi-Network’ (MN) TDOA, is proposed [6]. This approach allows expanding an existing network off the coast, by accounting for additional heterogeneous receivers (low-cost, temporarily deployed devices) carried by vessels, aircraft, buoys or balloons. In Figure 1, four different networks, which are not synchronised with each other, are considered. In this case, the traditional TDOA algorithm, which requires a high synchronisation among nodes, could not be exploited. In fact, even though the minimum condition for its application (at least three receivers in Line Of Sight (LOS)—Figure 2) is verified, the unknown time-offset between networks would lead to the divergence of the localisation algorithm [7]. On the contrary, the proposed MN TDOA would automatically compute the Intra-Network Bias (INB), guaranteeing the convergence of the positioning algorithm.

The proposed method is derived from that adopted by GNSS multi-constellation positioning and adapted to TDOA. A fundamental difference with respect to GNSS multi-constellation is the knowledge of the offset between the time scales. In fact, in the multi-constellation case, the offset is partially known, hence different strategies can be adopted to accommodate for such offset, while in the considered case the offset among the nodes is totally unknown. In [8,9,10], the authors proposed a comparison between different strategies to account for the inter-system bias; analogously, in this paper, a comparison between the proposed approach and the classical method (that does not compensate for the INB) is provided. The INB is composed of two terms: the difference between the network time scales and a device-dependent component. Whereas a continuous monitoring of the AIS time could allow accounting for the first term, the second term is extremely difficult to estimate.

This is why, to date, synchronisation issues between nodes represent the major limitation to the employment of TDOA-based localisation. In the case of AIS, the Time Of Arrival (TOA) estimate exploits the preamble of the message, yet different points of the preamble can be used to this end; moreover, each manufacturer can adopt its own strategy to compute this parameter. These different strategies may lead to unwanted offsets, when using a network made of receivers from multiple manufactures.

Several techniques has been proposed to solve this issue. For example, a network calibration procedure is described in [11] and, more recently, a method to automatically estimate both the unknown position and the time bias between nodes, by processing their periodically broadcasted signals, has been proposed in [12]. In [13], TDOA and Asynchronous TDOA (ATDOA) algorithms are evaluated for wide area multilateration systems, whereas Kim et al. [14] proposed a localisation system that simultaneously estimates the location of the client node and the local clock offsets. In [15], an ATDOA-based localisation algorithm that does not employ a synchronisation process is proposed.

The MN TDOA has been introduced for the first time in [6] and the impact of the INB on the method has been described in [16] for a static scenario. In [17], a set of basic localisation techniques for 3G mobile phones is presented; the analysis shows that TDOA-based localisation algorithms require either precisely synchronised clocks for all transmitters and receivers or a means to measure these time differences (otherwise, a 1 μs timing error leads to a 300 m position error). This is true for 3G receivers, which works with a much higher frequency and bandwidth than the AIS ones, and accordingly are more accurate. With this in mind, the benefit of the proposed method, which extends the traditional TDOA algorithm allowing to exploit measurements from non-synchronised devices (such as the prototype developed in [18]) should be evident. The approach is developed in the measurement domain; all the aspects relative to the signal domain (such as multipath, noise, Non Line Of Sight (NLOS) and interference in general) are accounted for by the distribution models of the error adopted to simulate different types of receivers.

Besides the AIS, there are other types of signals that can be exploited by the TDOA algorithm to localise targets. One of the most commonly used is the Global System for Mobile Communications (GSM), but, in principle, any radio signal is suitable for the radiolocation through TDOA approach. In this framework, the TDOA algorithm has been extensively investigated, starting from the methods for the estimation of the TDOA measurements [19]. Both TOA and TDOA measurement errors are analysed in [20]. The timing-difference concept is also used for the localisation of mobile stations, as shown in [21,22], whereas the author of [23] analysed the accuracy of cellular mobile station location estimation. Standardisation of mobile phone positioning for 3G systems has been proposed in [24], whereas, in [25], different techniques, including TDOA, are analysed to track and localise mobile phones. In [26], the authors addressed a joint source location and propagation speed estimation using TDOA; the approach is necessary in scenarios, such as underwater acoustic localisation, where the signal velocity is unknown. TDOA can also be used in very different domains, such as the event localisation technique in power systems based on Phasor Measurement Unit (PMU) [27] or the Compressed Sensing (CS) [28].

Alternative radiolocation techniques that are not based on TDOA can also be considered. The effectiveness of an indoor distance-estimation approach based on Orthogonal Frequency Division Multiplexed (OFDM) signals combined with the Zadoff–Chu sequences is demonstrated in [29,30]. The Simultaneous Localization And Mapping (SLAM) allows estimating the position of the transmitting target through its previous positions and the estimated cinematic features. This possibility, as well as the application of a tracking algorithm to a series of positions estimated with the MN TDOA approach, goes behind the scope of the paper and it is no further explored in the following.

This paper extends the results achieved in [6,16] by comparing classic TDOA and MN TDOA algorithms in a simulated maritime scenario. Both algorithms were tested against a moving target that follows a 4-h non-linear and discontinuous trajectory with variable speed and heading. Since the geometry of the network influences the localisation performance of the method, the simulation in a dynamic scenario gave the possibility to demonstrate the robustness of the approach in a multitude of different geometric conditions. The position of the target was estimated with a rate of 1 Hz, without any assumption on the offset between nodes (meaning that no propagation of the offset between epochs was performed, hence the impact of the offset-drift cannot be appreciated).

In the proposed approach, the positions of the randomly-dislocated additional nodes are a priori known. This is commonly true for static nodes, but it can be extended also to mobile nodes, given that they provide their GNSS position together with the measured TOA. Obviously, the GNSS positioning error should be accounted for by the proposed MN TDOA method but it would be lower than 10 m [31]. Methodologies similar to those proposed in [8,32,33] should be adopted whenever the mobile-nodes’ positions are not available. In addition, in this case, an additional error relative to the position of the receivers should be accounted for, degrading the overall performance of the proposed method.

It is worth remarking that, even though it was designed for AIS signals, the developed technique can be easily extended to all kind of Radio Frequency (RF) signals: Very high frequency (VHF) communications, frequency modulation (FM) radio, mobile phones, etc. Besides the validation of AIS positioning, a different application in the maritime domain could be the localisation of users in distress through signals of opportunity.

The following part of the paper is structured as follows. In Section 2, both the traditional and the enhanced TDOA algorithms are briefly presented, highlighting their differences. The simulation set-up is described in Section 3 and the relative results are shown in Section 4. Finally, Section 5 concludes the paper with key considerations.

## 2. TDOA Algorithms

In this section, the formulation of the traditional and the Multi-Networks TDOA (MN) algorithms are presented. The MN approach has been derived from the traditional TDOA algorithm, in order to account for the INB among the nodes of the receiving network In this sense, the MN can be seen as an extension of the traditional TDOA. A brief description of the traditional TDOA algorithm is provided in Section 2.1, and then its MN extension is presented in Section 2.2.

Both the MN and the traditional TDOA algorithms are based on TOA measurements, which can be modelled as [34,35,36,37,38]:(1)$$TOA=\frac{d}{c}+t_{0}^{t}+t_{0}^{r}+\epsilon_{TOA},$$
where $t_{0}^{t}$ is the transmitter timing error, $t_{0}^{r}$ is the receiver timing error, $\epsilon_{TOA}$ represents the residual error of the TOA measurement, *c* is the speed of light and *d* is the emitter–receiver distance:(2)$$d=\sqrt{{\left(x_{r}-x_{t}\right)}^{2}+{\left(y_{r}-y_{t}\right)}^{2}}$$
where $x_{r}$ and $y_{r}$ are the coordinates of the receiver, while $x_{t}$ and $y_{t}$ are the unknown coordinates of the transmitter. Usually, TOA measurements cannot be directly used for radiolocation, because of the unknown offset between the clock of the transmitter and that of the receiver. Such offset can make the positioning unreliable or even unfeasible. In addition, due to differences in electronic components and processing algorithms, the offset between receiver and emitter is device-dependent [18], hence very difficult to model. A continuous monitoring of the AIS network could take into account only the non-synchronisation between the network time scales and not the device-dependent offset; hence, by itself, it could result ineffective because of local effects. TOA measurements can be profitably used for positioning purposes only when the transmission time is perfectly known, such as for the case of the GNSS-based positioning [31,39,40,41]. When the time of transmission is not known, an approach which removes the effect of the unknown offset needs to be adopted. One of the possible solutions is the traditional TDOA algorithm (Figure 2).

### 2.1. Traditional TDOA Algorithm

In the traditional TDOA algorithm, the clock error (relative to the transmitter) is removed using the difference between two TOAs:(3)$$TDOA_{i,j}=TOA_{i}-TOA_{j},$$
where $TOA_{i}$ and $TOA_{j}$ are the TOAs estimated, respectively, by the *i*th and *j*th receivers. Substituting Equations (Equation 1) and (Equation 2) into Equation (Equation 3), the following expression for the TDOA measurements is obtained:(4)$$TDOA_{i,j}=\frac{\sqrt{{\left(x_{r}^{i}-x_{t}\right)}^{2}+{\left(y_{r}^{i}-y_{t}\right)}^{2}}-\sqrt{{\left(x_{r}^{j}-x_{t}\right)}^{2}+{\left(y_{r}^{j}-y_{t}\right)}^{2}}}{c}+\left(t_{0}^{r^{i}}-t_{0}^{r^{j}}\right)+\epsilon_{TDOA}$$

This new observable, namely TDOA, can be adopted for target localisation, using the approach described in [42,43]. It can be noted that, in Equation (Equation 4), the terms related to the transmitter timing error ($t_{0}^{t}$) disappear. Moreover, the term $t_{0}^{r^{i}}-t_{0}^{r^{j}}$ is neglected, hence Equation (Equation 4) reduces to [44]:(5)$$TDOA_{i,j}=\frac{d_{i}-d_{j}}{c}+\epsilon_{TDOA},$$
where $d_{i}$ and $d_{j}$ are the distances between the transmitter and the *i*th and *j*th receivers, respectively. Equation (Equation 5) has to be linearised as in [45]; hence, it is expanded in Taylor series considering the nominal position $p_{0}=\left[x_{0},y_{0}\right]$. Neglecting the terms higher than the first order, the linearised version of Equation (Equation 5) is given by:(6)$$TDOA=TDOA_{0}+\frac{\partial TDOA}{\partial x}{{|}_{x_{0}}\frac{x-x_{0}}{c}+\frac{\partial TDOA}{\partial y}|}_{y_{0}}\frac{y-y_{0}}{c},$$
where

$TDOA_{0}$ is the TDOA estimated considering the nominal position $p_{0}$;${\left.\frac{\partial TDOA}{\partial x}\right|}_{x_{0}}$ and ${\left.\frac{\partial TDOA}{\partial y}\right|}_{x_{0}}$ are the partial derivatives of the TDOA with respect to *x* and *y* orthogonal axis, respectively, evaluated in the nominal position $p_{0}$.

The two partial derivatives can be expressed as:(7)$$\begin{array}{c} \frac{\partial TDOA}{\partial x}{|}_{p_{0}}=\frac{x_{0}-x_{r}^{1}}{d_{0}^{1}}+\frac{x_{0}-x_{r}^{2}}{d_{0}^{2}}\\ \frac{\partial TDOA}{\partial y}{|}_{p_{0}}=\frac{y_{0}-y_{r}^{1}}{d_{0}^{1}}+\frac{y_{0}-y_{r}^{2}}{d_{0}^{2}}, \end{array}$$
where $\left[x_{r}^{1},y_{r}^{1}\right]$ and $\left[x_{r}^{1},y_{r}^{1}\right]$ are the the horizontal coordinates of the two receivers and $d_{0}^{1}$ and $d_{0}^{2}$ are the initial distances between the target and the receivers. Hence, replacing Equation (Equation 7) into Equation (Equation 6), one obtains:(8)$$TDOA-TDOA_{0}=\left(\frac{x_{0}-x_{r}^{1}}{d_{0}^{1}}+\frac{x_{0}-x_{r}^{2}}{d_{0}^{2}}\right)\cdot \frac{x-x_{0}}{c}+\left(\frac{y_{0}-y_{r}^{1}}{d_{0}^{1}}+\frac{y_{0}-y_{r}^{2}}{d_{0}^{2}}\right)\cdot \frac{y-y_{0}}{c}$$
where the term on the left side is the difference between measured and predicted TDOA. This term can be indicated as ΔTDOA.

By assuming:(9)$$\begin{array}{c} a=\frac{x_{0}-x_{r}^{1}}{d_{0}^{1}}+\frac{x_{0}-x_{r}^{2}}{d_{0}^{2}}\\ b=\frac{y_{0}-y_{r}^{1}}{d_{0}^{1}}+\frac{y_{0}-y_{r}^{2}}{d_{0}^{2}}\\ \Delta x=(x-x_{0})\\ \Delta y=(y-y_{0}) \end{array}$$
the compact form of Equation (Equation 8) is given by:(10)$$\Delta TDOA=\frac{1}{c}\cdot \left(a\Delta x+b\Delta y\right)+\epsilon_{TDOA}.$$
where Δx and Δy represent the corrections to the a priori position $p_{0}$ and *a* and *b* are the dimensionless multiplying factors along the two coordinates, which are the components of the design matrix connecting measurements and unknowns.

Considering a set of *k* measurements, the matrix notation of Equation (Equation 10) is:(11)$$\begin{array}{cc} \; & \Delta \mathrm{TDOA}=\left[\begin{array}{c} \delta TDOA_{1}\\ \delta TDOA_{2}\\ \vdots \\ \delta TDOA_{k} \end{array}\right]=\frac{1}{c}\left[\begin{array}{cc} a_{1} & b_{1}\\ a_{2} & b_{2}\\ \vdots & \vdots \\ a_{k} & b_{k} \end{array}\right]\left[\begin{array}{c} \Delta x\\ \Delta y \end{array}\right]+\epsilon_{TDOA} \end{array}$$
and it can be expressed in compact form as:(12)$$\Delta \mathrm{TDOA}=\frac{1}{c}H_{TDOA}Z+\epsilon_{TDOA}.$$

Here, ΔTDOA is the measurement vector, $H_{TDOA}$ is the design matrix connecting measurements and unknowns, and Z is the state vector (Δx,Δy).

The unknowns are estimated using the Weighted Least Squares (WLS), even though different estimation method could be exploited. After the estimation process, the emitter position is given by:(13)$$p=p_{0}+Z.$$
where p at epoch $t_{1}$ is considered as nominal position at epoch $t_{2}$ and so on.

### 2.2. Multi-Network TDOA Algorithm

In the traditional algorithm, the term $t_{0}^{r^{i}}-t_{0}^{r^{j}}$ is neglected; this assumption is valid only when receivers with the same characteristics are used or when their offset can be reasonably modelled and accounted for. Usually, this is not the case (especially when using different networks of receivers) and such offset is unknown, potentially leading to the divergence of the localisation algorithm. To fill this gap, an alternative measurements model has been developed. TDOAs measurements obtained from dissimilar nodes (i.e., receivers with different hardware and/or software characteristics) can be modelled as:(14)$$TDOA_{i,j}=\frac{d_{i}-d_{j}}{c}+INB+\epsilon_{TDOA},$$

Equation (Equation 14) is very similar to Equation (Equation 5), but there is an additional term, namely ’INB’, accounting for the offset between the different clocks of the nodes. Similar to Equation (Equation 5), Equation (Equation 14) is not linear in the unknowns, hence a linearisation process similar to the one described in the previous section has to be performed. Specifically, Equation (Equation 14) is expanded in Taylor series, considering as initial position $p_{0,MN}=\left[x_{0},y_{0},INB_{0}\right]$ (with $INB_{0}$ assumed equal to zero):(15)$$TDOA=TDOA_{0}+\frac{\partial TDOA}{\partial x}{|}_{p_{0,MN}}\frac{x-x_{0}}{c}+\frac{\partial TDOA}{\partial y}{{|}_{p_{0,MN}}\frac{y-y_{0}}{c}+\frac{\partial TDOA}{\partial INB}|}_{p_{0,MN}}\left(INB-INB_{0}\right).$$

Applying the same manipulations described in the previous section, Equation (Equation 15) becomes:(16)$$\Delta TDOA=a\frac{\Delta x}{c}+b\frac{\Delta y}{c}+f\Delta INB,$$
where *a* and *b* are the same defined as in Equation (Equation 9); *f* is zero when node with similar characteristics are considered and one when measurements from non-synchronised nodes are used. The matrix form of Equation (Equation 16) is the following: (17)$$\Delta \mathrm{TDOA}=\left[\begin{array}{c} \delta TDOA_{1}\\ \delta TDOA_{2}\\ \vdots \\ \delta TDOA_{k} \end{array}\right]=\left[\begin{array}{ccc} a_{1} & b_{1} & 0\\ a_{2} & b_{2} & 0\\ \vdots & \vdots & \vdots \\ a_{k} & b_{k} & 1 \end{array}\right]\left[\begin{array}{c} \Delta x/c\\ \Delta y/c\\ \Delta INB \end{array}\right]+\epsilon_{TDOA}$$

Equation (Equation 17) shows an example of the design matrix for the proposed MN TDOA method. The zeros in the last column are relative to TDOA measurements given by the difference of TOAs between similar receivers. On the other hand, when TOAs from receivers with different characteristics are used, then the element of the last column is set to 1. The design matrix shown in Equation (Equation 17) is valid for the particular case where the INB is the same for all the additional receivers. If such assumption is not valid, the design matrix can be expanded adding additional columns for the estimation of the different INBs. In this case, also the state vector will contain additional elements ΔINBs, which allow estimating the INBs for the different types of receiver.

The main differences between the traditional and the proposed MN algorithms are summarised in Table 1. A flow chart showing the proposed MN TDOA algorithm applied to AIS position validation is proposed in Figure 3. The differences between this algorithm and the traditional TDOA algorithm are in the measurement domain (green-dashed box on the right) and consist in the automatic compensation of the INB through the estimation of the unknowns ΔINBs. This allows using different networks of not-synchronised receivers. In the final part of the flow chart, the position of the target, which is self-provided through AIS, is matched with that estimated through the proposed method.

### 2.3. Geometry

Localisation errors are strongly related to the measurement errors: usually the error in the position domain is obtained multiplying the measurement error by a factor relative to the geometry of the receivers [39,46]. This geometric factor can be obtained from:(18)$$DOP_{matrix}=\sigma_{0}^{2}\cdot A$$
where $\sigma_{0}$ is the a priori variance of the measurements and *A* is:(19)$$A={\left(H^{T}H\right)}^{-1}$$

The design matrix *H* is built as discussed in the previous sections. It can be observed that, *A* is only a function of the relative transmitter–receiver positions; hence, it represents the geometry of the system and it is usually referred to as Dilution Of Precision (DOP) matrix. *A* is an m×m matrix, where *m* is the number of unknowns to estimate. Each element on the main diagonal of *A* represents the geometric factor of the different unknowns; for example in the case of the MN algorithm, three unknowns have to be estimated, namely *x*, *y* and INB, hence the 3×3 DOP matrix has the following elements on the main diagonal: xDOP (i.e., East DOP (EDOP)), yDOP (i.e., North DOP (NDOP)) and INBDOP, respectively [6].

Equation (Equation 18) is only valid in the case of equal noise on all TDOA measurements, a more complex DOP can be computed (WDOP) considering a weighting matrix *W* based on SNR. The use of such a matrix may improve the performance of the algorithm. However, the DOP matrix stills represent the geometry of the system.

## 3. Simulation Set Up

In this section, the simulated tests used to validate the MN TDOA algorithm are described.

A kinematic test was simulated, including trajectories with different speeds, headings and static points. The overall length of the test was about 4 h, corresponding to 16,503 epochs, by assuming a data rate of 1 Hz. The schematic representation of the test is shown in Figure 4.

In the first part of its navigation, the vessel moves with a constant speed of about 13.5 knots (approximately 25 km/h) and a heading of $45^{\circ }$ (NE direction); then, a stop of 30 min is simulated. After the stop, the vessel moves again along a circular path with a constant tangential velocity of 8.5 knots (15.7 km/h). The radius of the path is 3.1 NM (5.7 km), and its length 8.5 NM (15.7 km). After the circular path, the vessel stops for 10 min, andthen it moves again for 16 min and 40 s, with speed 19.5 knots (36 km/h) and heading $286^{\circ }$ ( south-southeast (SSE) direction), covering a distance of 5.4 NM (10 km). The total length of the path is more than 40.5 NM (about 75 km). In Table 2, the parameters describing the simulated path are summarised.

Two types of receivers, characterised by different performances, were simulated: low-cost and high-end. For both types of receivers, the measurement errors were modelled as Gaussian noise, but with different characteristics, as described in [6] and summarised in Table 3. The distribution of the TDOA errors, considering a network composed by three synchronised nodes and one non-synchronous device, is shown in Figure 5. It is evident how the Gaussian bells relative to TDOA errors that include receiver number 4 appear larger than the others. It can also be noted that the error distribution shown in this figure is consistent with that reported in [47], where the noise analysis was carried out on real data collected by different AIS stations belonging to the Italian Coast Guard network. In fact, the parameters of the errors distribution in Table 3 have been chosen to simulate realistic devices [47].

The test was performed considering a baseline scenario, composed by three high-end ground-based receivers, and a variable number of additional low-cost receivers, ranging from 1 to 50 (see Figure 6). The INB is considered constant and equals to 0.5 ms; the offset is assumed to be the same for all the additional receiver. The assumption on the constant value of the INB does not affect the estimation process; because the estimation is performed with a snapshot approach, a single epoch is considered for each estimation.

The simulated position of the additional receivers was chosen randomly, but respecting a simple geographical constraints: no additional receivers are placed on land. Using this approach, the low-cost receivers appear as they were deployed by a ship operating in that area. This could be a typical scenario where the the small coastline receivers network is temporarily expanded in order to improve the detection capability in that given area.

### Cramer–Rao Bound Results

In this section, the results of the Cramer Rao Bound (CRB) analysis are presented. The CRB provides a lower bound for the variance of an unbiased estimator; the variance of the estimator has to satisfy the following criteria [13]:(20)$$var\left(\theta \right)\geq [I^{-1}\left(\theta \right)]$$
where var(θ) is the variance of the estimator and *I* is the Fisher information matrix.
In the case of Gaussian error distribution, the Fisher information matrix can be written as [13,48]:(21)$$I=H^{T}\cdot Q\cdot H$$
where *H* is the design matrix defined in Equation (Equation 17) and *Q* is the covariance matrix given by Equation (Equation 22), as described in [13,48].
(22)$$Q=\left[\begin{array}{ccccc} 1 & 0.5 & \cdots & 0.5 & 0.5\\ 0.5 & 1 & \cdots & 0.5 & 0.5\\ \vdots & \vdots & \ddots & \vdots & \vdots \\ 0.5 & 0.5 & \cdots & 1 & 0.5\\ 0.5 & 0.5 & \cdots & 0.5 & 1 \end{array}\right]$$

Using the previous equations, the CRB for the two coordinates is computed as a function of the number of additional receivers. The results are shown in Figure 7. In the figure, it can be noted a strong reduction of the CRB passing from 1 to 20 additional receivers. An asymptotic behaviour of the three components can be noted after 20 additional receivers. As expected, the CRB relative to the INB is more affected from the introduction of additional receivers.

## 4. Results

In this section, the results obtained using the proposed MN TDOA algorithm are presented. The performance of traditional TDOA and MN algorithms are compared in terms of both geometry and horizontal position error.

### 4.1. Geometry Results

The geometry conditions of the test are shown in Figure 8, where the mean Horizontal DOP (HDOP) is shown as a function of the number of additional receivers for both the traditional and the MN TDOA. The DOP matrix defined in Equation (Equation 18) is composed of EDOP, NDOP and INBDOP. The HDOP is obtained as combination of EDOP and NDOP.

Figure 9 shows the mean DOP obtained using the MN approach and its break down by components (EDOP, NDOP and INBDOP, which is referred to as TDOP) as a function of the number of additional receivers. The three DOPs have similar asymptotic behaviour: basically no relevant benefits can be appreciated adding more than 20 additional receivers. In the first part of the graph (with 1–5 additional receivers), the INBDOP is lower than EDOP and NDOP, while, in the second part, the INBDOP is higher than the other components. The geometric results are strictly linked to the location of the additional receivers with respect to that of the target-vessel.

The geometry conditions are strongly improved when passing from one to five additional receivers: for the traditional algorithm, the HDOP is reduced by a factor of six, going from 2.4 to 0.4, whereas, for the MN case, it is reduced by a factor of three, passing from 1.6 to 0.6. The fact that, for a low number of additional receivers, the proposed method underperforms the traditional one in terms of HDOP is due to the different number of synchronous and non-synchronous TDOA measures, as reported in Table 4 and further explained at the end of this section. However, it can be noted that both algorithms converge to the same HDOP value.

The number of synchronous and non-synchronous TDOA measurements for the different configurations of the receivers network is given in Table 4. The table further supports what is presented in Figure 8, where for a number of additional receivers lower than 6, the relative number of synchronous TDOA measurements is lower than that of non-synchronous. As noted in the table, the number of non-synchronous observable is higher than the synchronous ones considering configurations from 3+1 to 3+5. This explains why until five additional receivers the INBDOP is higher than the positioning-related DOPs (Figure 9).

Figure 10 shows the HDOP computed on different points of the reference trajectory: the upper boxes are relative to the MN cases, whereas the lower boxes are relative to the traditional TDOA. For both algorithms, four configurations are considered: each one including the three ground-based base stations and an increasing number of additional receivers (two (5 Rx), five (8 Rx), ten (13 Rx) and twenty (23 Rx)). From the results in the figure, it emerges that for both algorithms the largest improvements are obtained passing from two to five additional receivers). A small improvement can be noted passing from five to ten additional receivers (mainly for the MN case), and finally no particular benefits can be appreciated passing from ten to twenty additional receivers. This confirms the results shown in Figure 8. From the comparison between the boxes relative to the same configurations (i.e., comparing the results vertically), it emerges that the HDOP values obtained using the MN approach are higher than the ones obtained using the traditional algorithm. For example, comparing the two figures on the left, it can be noted that the trajectory relative to the MN case has more yellowish and orangish sections. At first glance, one could say that the proposed approach shows lower performance than the traditional one. This is true in terms of HDOP and the reason is the necessity to estimate an additional unknown in the case of the MN algorithm.

The improvements in the localisation domain are linked to the location of the additional devices. In [49], the authors provided a mathematical demonstration of the fact that adding an additional receiver improves DOPs. From the reference, it can be noted that the geometry is improved even if a receiver is placed in a non-optimal location. The results presented and the limit of 20 additional receivers for the performance improvement are valid only in the specific case here considered, where the additional receivers were positioned randomly. In fact, the improvements would be higher if an optimisation of the location of the additional devices is adopted.

### 4.2. Tracking Results

To analyse the effects of the introduction of additional receivers in the position domain, the estimated trajectories using the MN and the traditional algorithms are shown in Figure 11 and Figure 12, respectively. It is important to remark that within this section no tracking algorithm is applied after the estimation of the position of the transmitting target with the proposed approach. In fact, the term ‘tracking’ is used to indicate a repeated estimation of the target position along its trajectory.

For sake of visualisation, only five configurations are considered in both the proposed figures. Specifically, the configurations considering 2, 5, 10, 20 and 50 additional receiver are shown. The benefits of the inclusion of additional receivers for the MN case are evident in Figure 11, where it can be noted that the estimation noise is clearly reduced by enhancing the number of additional receivers. The main improvements are obtained passing from 2 to 5 additional receivers. However all the trajectories properly represents the reference trajectory (black line) and no relevant bias can be noted. In Figure 12, the trajectories obtained with the traditional algorithm are shown. In addition, in this case, the benefit of the introduction of additional receivers are evident, but, contrarily from the previous case, a bias can be clearly noted between the estimated trajectories and the reference one. The bias is due to the unknown offset between the not synchronised nodes of the network, which is not taken into account by the traditional algorithm.

To directly compare the trajectory obtained using the two algorithms, the estimated solutions are shown in Figure 13. In each box, a different configuration of the receiving network is assumed. Beside the baseline network of three high-end base stations, the following number of low-cost receivers is assumed for the relative box: 2 in the upper-left box, 5 in the upper-right box, 10 in the bottom-left box and 20 in the bottom-right box. In the figure, it is evident that the MN algorithm outperforms the traditional one. The estimates using the MN algorithm (blue lines) are centred on the reference trajectory (black line), whereas an evident bias can be noted when the traditional algorithm is adopted.

In Figure 14, the horizontal errors and the CRB as a function of the epoch are shown. The horizontal error for each configuration and the horizontal CRB are computed in each point of the trajectory (correspondent to a given epoch) in order to account for the different geometric conditions. In the figure, it can be noted that the introduction of additional receivers allows reducing the horizontal error. This effect is more evident in the second part of the test, when the vessel moves away from the baseline receivers. For the configuration with two and five additional receivers (violet and orange lines respectively), the positioning error grows when the epoch goes from zero to some 6000 s; this is mainly due to the geometric condition along the trajectory. This effects is not evident considering ten and twenty additional receivers, because such number of additional devices allows a geometric improvement along all the trajectory, as also shown in Figure 10. A similar behaviour can be observed for the CRB, which is lower than the computed errors.

To further investigate the benefits of the introduction of additional receivers, the Cumulative Distribution Functions (CDFs) of the horizontal position errors are shown in Figure 15, which allows a performance comparison between the two algorithms. For both approaches, the benefits due to the inclusion of additional receivers, as observed in the previous figures, are confirmed. Considering that each colour is associated to a different configuration of the receivers network, by comparing the two algorithms, it clearly emerges that the MN (continuous lines) outperforms the traditional (dashed lines) TDOA. No bias can be appreciated for the cases of the MN configurations, whereas a bias can be noted for the traditional approach. Specifically, the dashed lines are flat in the bottom-left part of the graph, showing a bias between 1 and 3 km (note that the values on the x-axis are expressed in tens of kilometres).

## 5. Conclusions

An extension of the traditional TDOA radiolocation algorithm, namely MN TDOA, was implemented and tested. The proposed approach allows exploiting measurements from non-synchronous devices. Thanks to this feature, an existing network of receivers can be easily expanded, including additional receivers with different and unknown characteristics, to cover a larger area. This approach can be adopted in several applications, such as the possibility to verify the veracity of a ship position provided through AIS.

The paper compares the radiolocation performance of the novel method with that obtained through the traditional TDOA approach. The simulated scenario includes a moving vessel (target) with non-uniform trajectory and speed, together with a non-synchronised receiving network made by a baseline configuration of three high-end coastal receivers and a varying number of additional low-cost receivers randomly dislocated at sea. The performance of both high-end and low-cost receivers are simulated in line with that of real receivers.

The main limitations of the proposed method are the following:a priori knowledge of the position of the additional stations and its accuracy;necessity to know whether or not the nodes are synchronised to proper set the design matrix; andincreased complexity of the algorithm and relative computational load.

The MN TDOA algorithm allows exploiting data from non-synchronous nodes by automatically computing the INB; on the contrary, the traditional TDOA applied to a non-synchronous network of receivers leads to an unbearable positioning bias, due to the unknown offset between not synchronised nodes. This is evident by matching Figure 11 with Figure 12 and again with Figure 13 where the MN TDOA outperforms the classical approach for any configuration of the additional receivers. Moreover, when comparing the two methods in the simulated scenario (i.e., under the adopted hypothesis on TOA error for both high-end and low-cost receivers and assuming the given target-trajectory and receivers-position), the following considerations can be made:The MN TDOA shows an HDOP equal to: 0.6 with 5 additional receivers; 0.2 with 10 additional receivers; and less than 0.1 with 20 additional receivers. No relevant benefits can be appreciated by further increasing the number of additional low-cost receivers.Considering 10 or more additional receivers, the horizontal error is lower (90% of the times) than 1 km for the MN TDOA and than 6.5 km for the traditional TDOA.

Although a 1-km error may appear huge, it is still acceptable in many applications, such as the proposed validation of AIS position. Moreover, the horizontal error can be further reduced by relying on higher performance receivers and by optimising their displacement within the area of interest. It is also worth remarking that no tracking algorithms have been applied and that with the expression ‘*estimated trajectory*’ the authors intend a series of consecutive estimated position along the vessel path. Indeed, any tracking algorithm (i.e., one based on a Kalman Filter) applied to the positions estimated with the MN TDOA would increase the accuracy of the final trajectory. Nevertheless, this is out of the scope of the paper and it could be eventually investigated in a different context. On the contrary, the comparison of the proposed method with different positioning strategies that account for the INB is a very relevant topic to be addressed in future works. Finally, the eventual availability of real data from a network including both low-cost and high performance AIS receivers will allow the authors to validate the proposed approach in a real scenario.

## Figures and Tables

**Figure 1 sensors-20-01842-f001:**
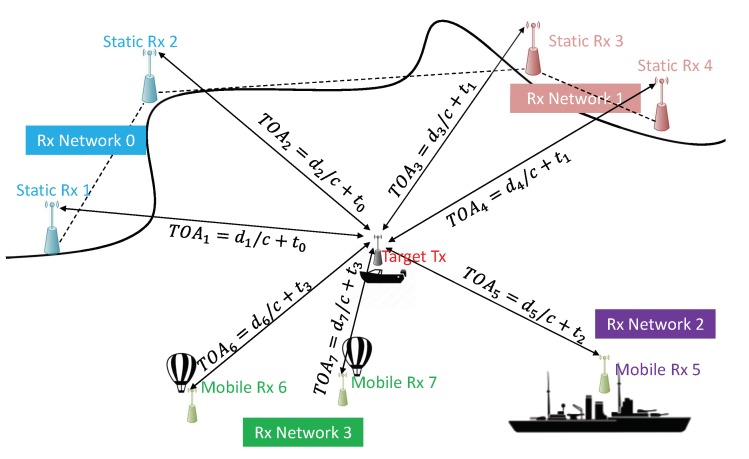
Typical scenario for Multi Network Time Difference of Arrival (TDOA). The four receiving networks (which include heterogeneous receivers) are not synchronised with each other ($t_{0}\neq t_{1}\neq t_{2}\neq t_{3}$). In this case, the traditional TDOA algorithm would lead to unbearable positioning errors.

**Figure 2 sensors-20-01842-f002:**
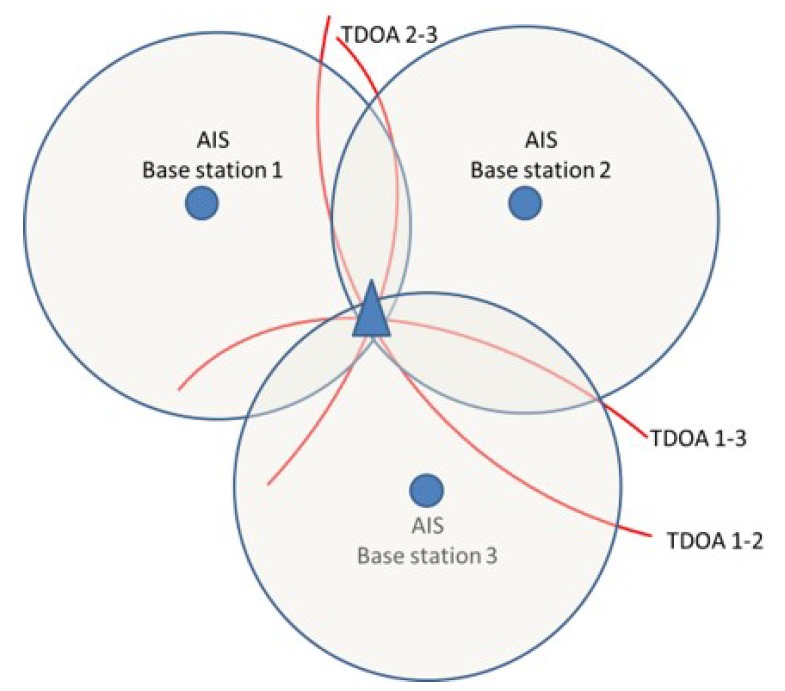
Basic triangulation with TDOA positioning. The three receiving nodes (blue circles) measure the TOA of the signal transmitted by the target (blue triangle). Exploiting the difference of TOAs, the position of the target is estimated.

**Figure 3 sensors-20-01842-f003:**
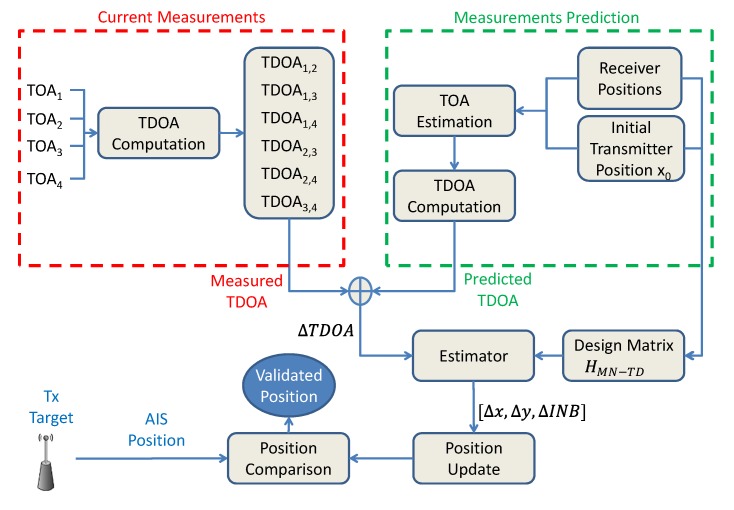
Flow chart of the proposed MN TDOA algorithm applied to AIS position validation. The novelty is the automatic compensation of the INB (through the estimation of the unknowns ΔINBs), which allows using different networks of not-synchronised receivers.

**Figure 4 sensors-20-01842-f004:**
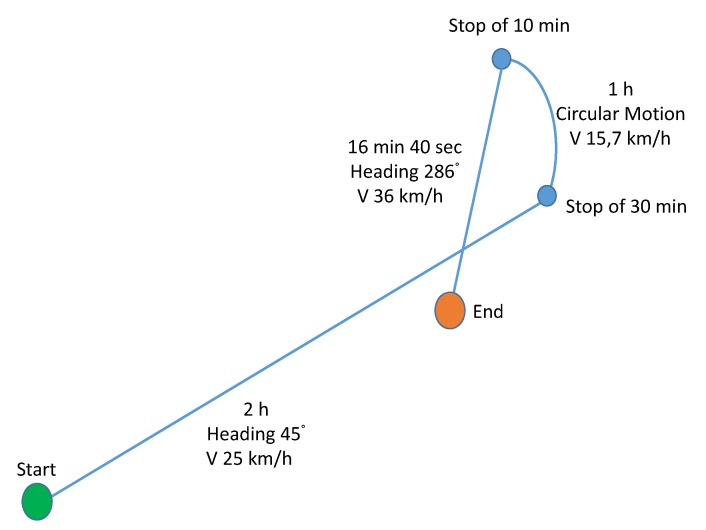
Schematic representation of the simulated test for the proposed radiolocation algorithm. The target ship (transmitter) departs from point ’Start’ and arrives to point ’End’ moving with the described speeds and headings.

**Figure 5 sensors-20-01842-f005:**
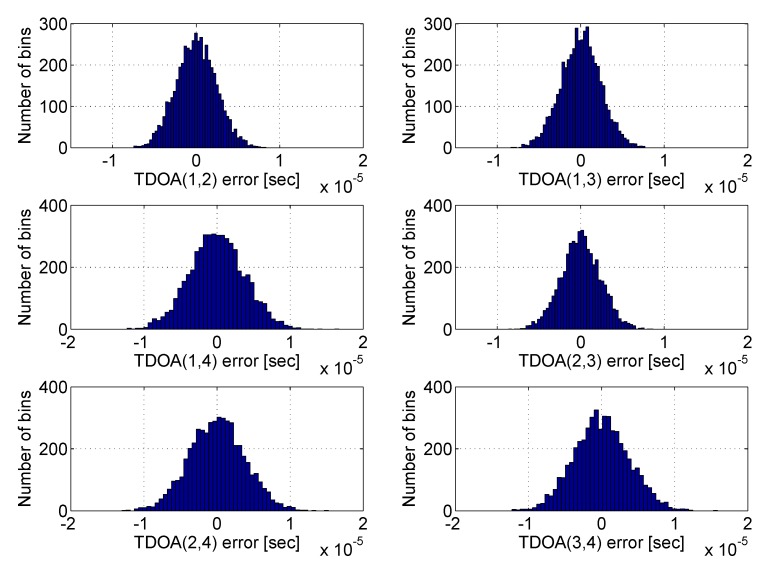
TDOA error distribution between couples of receivers. Three synchronised receivers (1–3) and one non-synchronous receiver (4) are considered. The Gaussian bells relative to TDOA errors that include receiver number 4 appear larger than the others.

**Figure 6 sensors-20-01842-f006:**
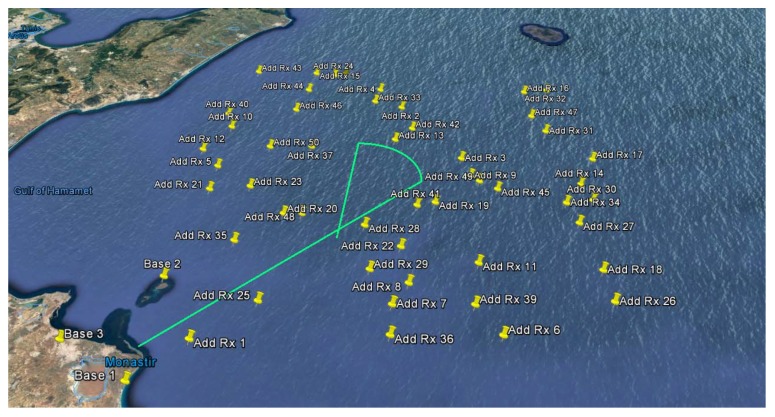
Position of the three high-end coastal stations (*Base #*) and of the additional low-cost receivers (*Add Rx #*). The simulated trajectory of the target vessel, leaving the Tunisian coast and heading toward Pantelleria island, is represented by a cyan line.

**Figure 7 sensors-20-01842-f007:**
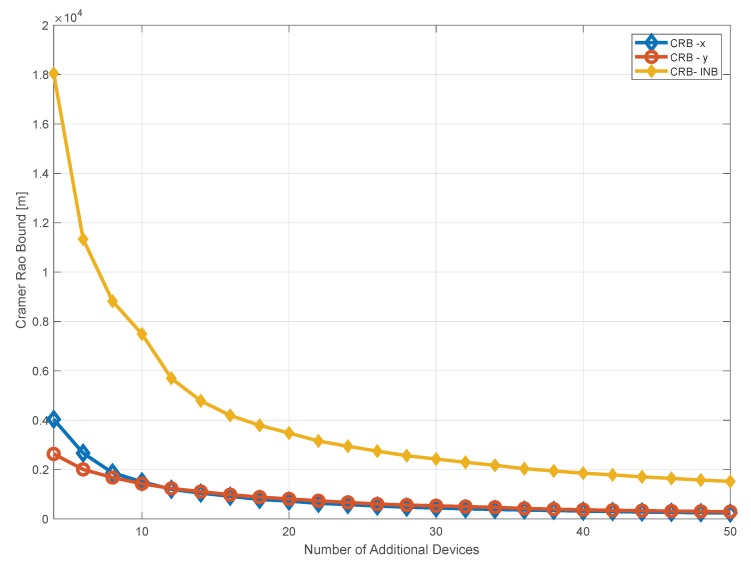
CRB as a function of the number of additional receivers for the three unknowns: the two coordinates *x*, *y* (blue and red, respectively) and the INB (yellow).

**Figure 8 sensors-20-01842-f008:**
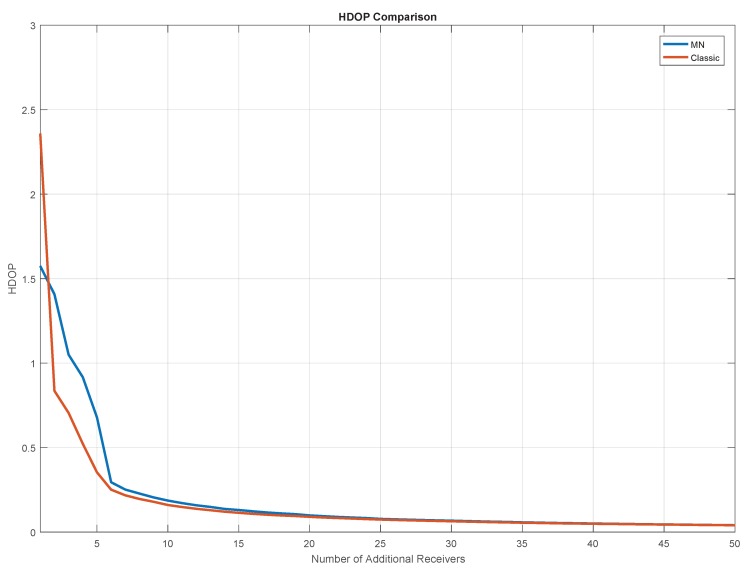
Comparison between traditional (red) and MN TDOA (blue) algorithms in terms of Horizontal DOP (HDOP) as a function of the number of additional receivers.

**Figure 9 sensors-20-01842-f009:**
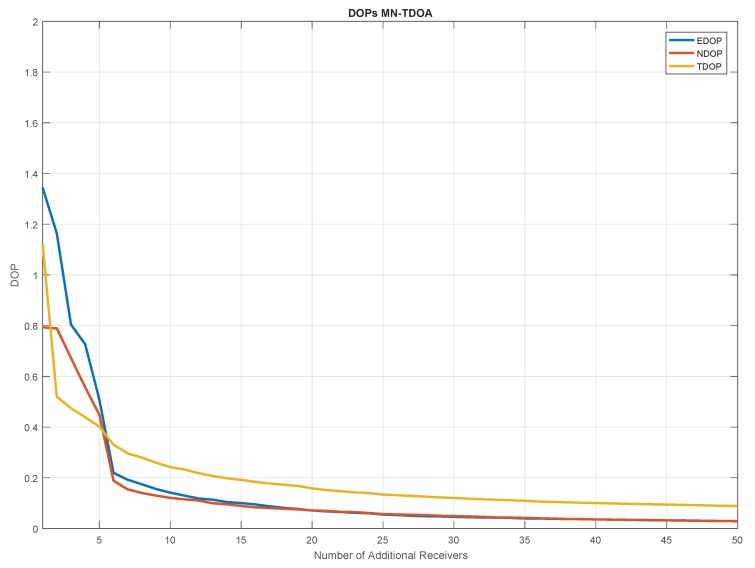
Dilution Of Precisions (DOPs) as a function of the number of additional receivers for the MN TDOA algorithm. East, north and time (INB) components of the DOP are represented, respectively, in blue, red and yellow.

**Figure 10 sensors-20-01842-f010:**
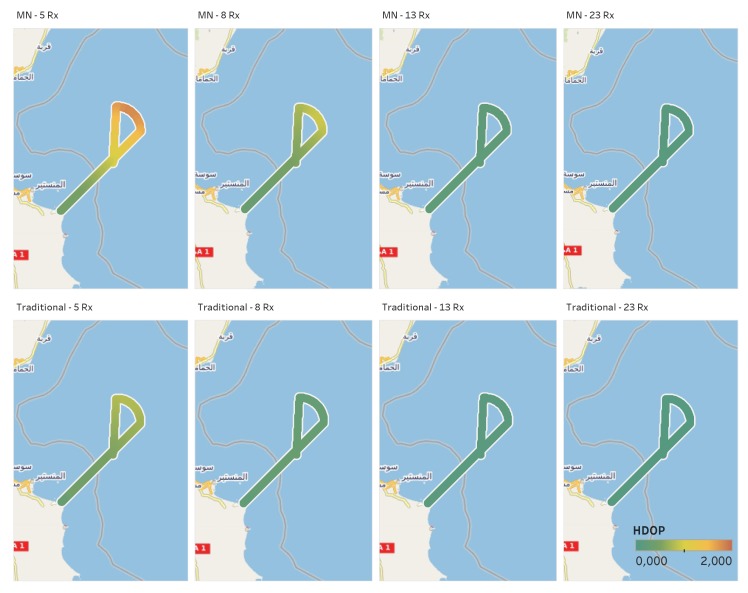
HDOP (colour-scale) along the vessel trajectory for the MN (top) and the traditional (bottom) TDOA in four different configurations of the receivers network (increasing number of additional receivers from left to right: BL+5Rx, BL+8Rx, BL+13Rx and BL+23Rx).

**Figure 11 sensors-20-01842-f011:**
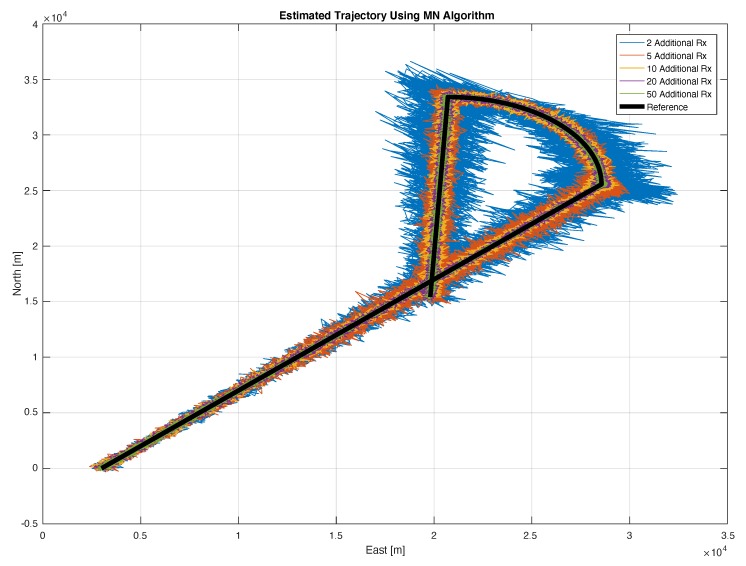
Trajectory of the simulated target (bold black line) and trajectories estimated with the MN TDOA algorithm (different colours represent estimation of the trajectory carried out with a different number of additional low-cost receivers: from 2 to 50).

**Figure 12 sensors-20-01842-f012:**
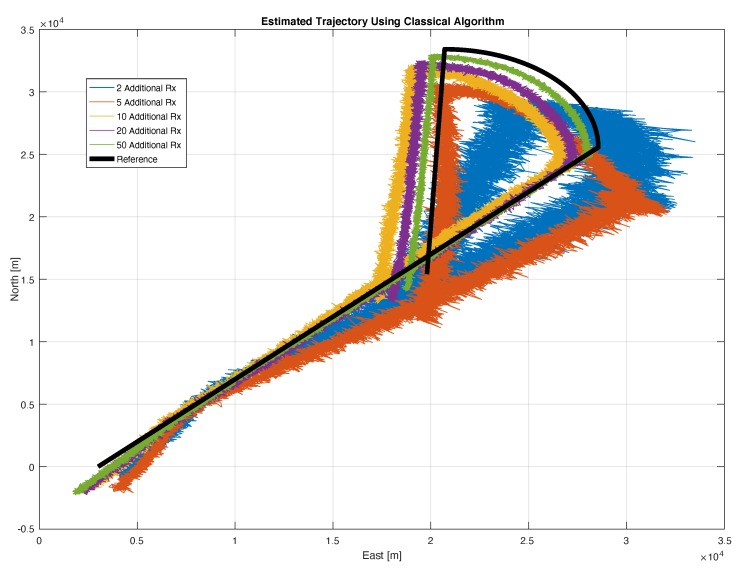
Trajectory of the simulated target (bold black line) and trajectories estimated with the traditional TDOA algorithm (different colours represent estimation of the trajectory carried out with a different number of additional low-cost receivers: from 2 to 50).

**Figure 13 sensors-20-01842-f013:**
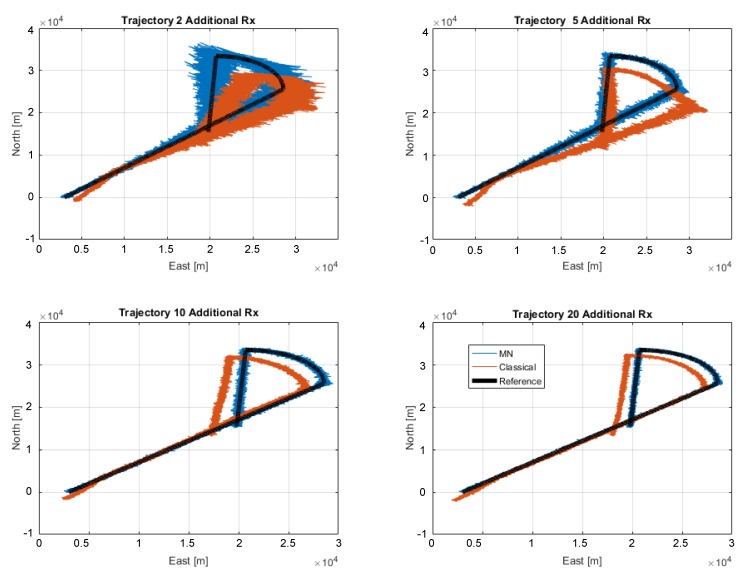
Comparison between the actual trajectory (black) and that estimated with the traditional (red) and with the MN (blue) algorithms. Four configurations of the receivers network are considered by adding to the the baseline Rx, respectively: 2 (top-left), 5 (top-right), 10 (bottom-left) and 20 (bottom-right) additional Rx.

**Figure 14 sensors-20-01842-f014:**
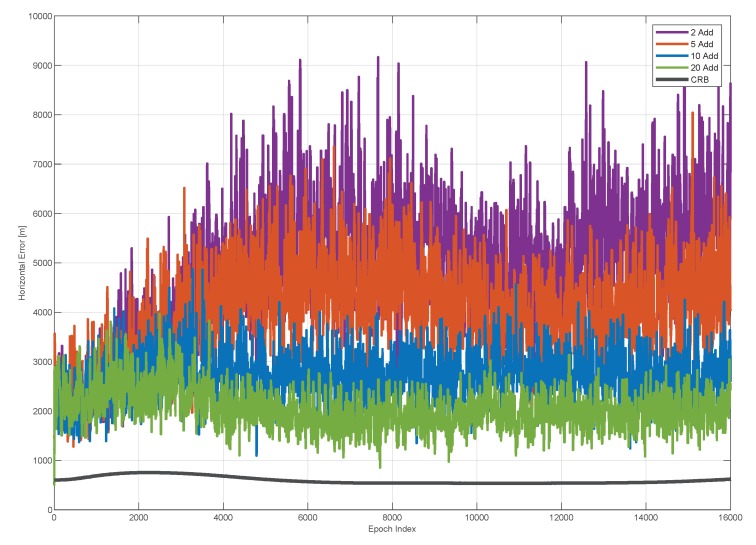
Horizontal Error as a function of the time epoch for four different configurations of the receiving network: BL+2Rx (purple), BL+5Rx (red), BL+10Rx (blue) and BL+20Rx (green). The horizontal error for each configuration and the horizontal CRB (black) are computed in each point of the trajectory in order to account for the different geometric conditions.

**Figure 15 sensors-20-01842-f015:**
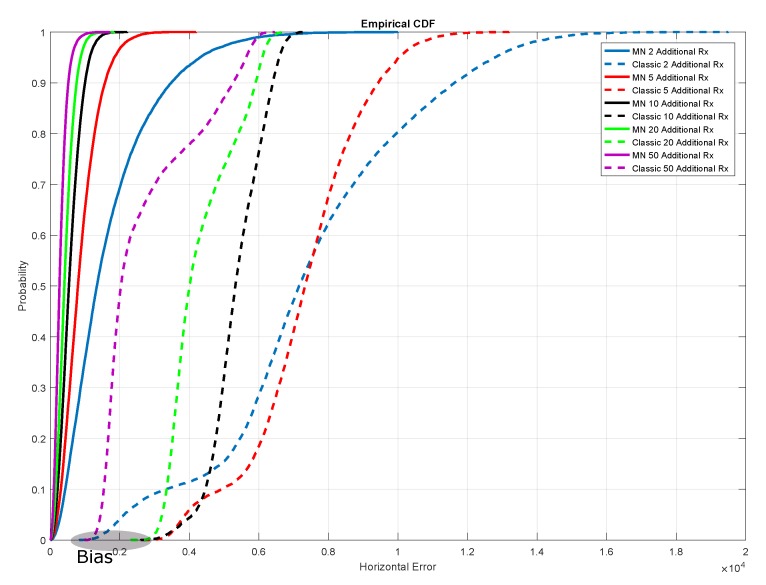
CDFs of the horizontal position errors for the traditional (sketched lines) and the MN TDOA (continuous lines) algorithms. Five configurations of the receiving network are considered: BL+2Rx (blue), BL+5Rx (red), BL+10Rx (black), BL+20Rx (green) and BL+50Rx (purple). The horizontal error on the x-axis is expressed in metres (multiplied for a factor 10,000). Note that, for each configuration, the error was computed at each point of the trajectory in order to account for the different geometric conditions.

**Table 1 sensors-20-01842-t001:** Main differences between the Traditional and the proposed MN TDOA algorithms.

Traditional TDOA	Parameters	MN TDOA
$TDOA_{i,j}=\frac{d_{i}}{c}-\frac{d_{j}}{c}+\epsilon_{TDOA}$	Measurement Model	$TDOA_{i,j}=\frac{d_{i}}{c}-\frac{d_{j}}{c}+INB+\epsilon_{TDOA}$
$H_{TDOA}=\left[a,b\right]$	Design Matrix	$H_{TDOA}=\left[a,b,f\right]$
$\Delta p=\left[\Delta x,\Delta y\right]$	State Vector	$\Delta p=\left[\Delta x,\Delta y,\Delta INB\right]$

**Table 2 sensors-20-01842-t002:** Parameters of the simulated path of the ship, decomposed by different kinematics and trajectories.

Phase	Type of Motion	Heading	Average Speed (km/h)	Duration	Distance (km)
1	Linear	$45^{\circ }$	25	2 h	50
2	Static	N.A.	0	30 min	0
3	Circular	N.A.	15.7	1 h	15.7
4	Static	N.A.	0	10 min	0
5	Linear	$286^{\circ }$	36	16 min 40 s	10

**Table 3 sensors-20-01842-t003:** TOA error parameters for the two types of simulated receivers.

Receiver Type	σ (ms)	*Mean* (ms)
High-end	0.0017	$3.336\times 10^{-5}$
Low-cost	0.0034	$3.336\times 10^{-4}$

**Table 4 sensors-20-01842-t004:** Number of synchronous and non-synchronous Time Difference Of Arrival (TDOA) measurements for the different configurations of the receivers network.

Rx Network Composition	Synchronous TDOA Measures	Non-Synchronous TDOA Measures
3 base + 1 add.	3	3
3 base + 2 add.	4	5
3 base + 3 add.	6	9
3 base + 4 add.	9	12
3 base + 5 add.	13	15
3 base + 6 add.	18	18
3 base + 7 add.	24	21

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
