# Peer review of "Multi-Network Asynchronous TDOA Algorithm Test in a Simulated Maritime Scenario"

_sensors, 2020, doi:10.3390/s20071842_

Round 1

Reviewer 1 Report

This paper deals with a method, which is derived from that adopted by GNSS multi-constellation positioning and adapted to Time Difference Of Arrival (TDOA). A fundamental difference with respect to Global Navigation Satellite System (GNSS) multi-constellation is the knowledge of the offset between the time scales. In the multi-constellation case, and the offset is partially known, hence different strategies could be evaluated and  adopted by the authors to accommodate for such offset, while in the considered case the offset among the nodes was totally unknown. Besides, the authors carried out a comparison between the proposed approach and a classical method, and they have shown why the traditional method may result ineffective.

However, it was expected a brief discussion and a better formalism about the used terms and its related processes, i.e., for instance, a better description of the algorithms, architecture and infrastructure to allows such a better scenarios.

In such a context, pseudo-codes algorithm, flow charts or even a UML (Unified Modeling Language), should be included in the manuscript with the purpose to clarify the visualization of the system, roles, actions, artifacts or classes, in order to better understand the authors ‘ideas.

The article alerts the readers to the possible benefits of the TDOA. Nevertheless, the introduction needs revision and a better contextualization in relation the state of the art related to such subject. 

For the equations all variables should be described and verified in terms of their physical dimensions. 

Practically all Figures should be better explained. For instance, in Figure14 the authors have shown the cumulative distribution functions (CDFs) of the horizontal position errors for the traditional (sketched lines) and the MN TDOA (continuous lines) algorithms results. But it was not clear how they take into account such boundary conditions to carry out such evaluation.

The conclusions of the article are quite superficial, and the article does not highlight the main contributions and the main results obtained in relation to those already available in the literature. Furthermore, there are periods that should be revisited. For example, the period “The benefits of the MN TDOA applied to a network of non-synchronised receivers are evident when compared to the traditional approach”, the authors should presented, as part of their conclusions, why the benefits are evident and how they have been proved.

Reviewer 2 Report

The authors test a proposed algorithm (named Multi-network TDOA algorithm) in a simultaed matitime scenario. The paper must be improved in the experimental presentation.

Here my considerations:

-The authors claim to overcome the limits of the classical TDOA implementation, by using heterogeneous low cost receivers carried by mobile nodes. This is a very interesting. The assumption made for the simulation is that the mobile nodes are in a priori known positions, which is is not realistic in the proposed reference scenario. Please add some considerations about the limit of the proposed approach in a real scenario. How do the meothodology in [22]-[24] affect the final results?

-The short anticipation of the results (the reference to section 4 and to Figg. 7 and 8) in the introduction, makes comprehension of the text difficult.

- The authors should cite recently published works on TDOA. I think the authors should cite the following papers to improve their literature review: https://doi.org/10.1109/TCOMM.2020.2973961, https://doi.org/10.1109/LCOMM.2020.2968434, https://doi.org/10.1109/WiMOB.2017.8115766, https://doi.org/10.1109/TPWRS.2019.2927613, https://doi.org/10.1109/TVT.2020.2968118, https://doi.org/10.1109/NEWCAS.2017.8010174, https://doi.org/10.1109/TAES.2020.2966095

- Check the caption of the fig.2

- check the formulas impagination

- In the simulation setup check the convertions between meter and nautical mile. (3.1 NM -- 5.7km). Fix the the units of measurement for the nautical mile, it could be nmi of NM.

- check the simulated test proposed. There is a mismatch between the description at line 210 (4.5 hours) and the data in Fig. 3 (almost 4 hours)

- check the axis scale of figg. 10-12. Also in this case they seem to not fit with the data in fig. 3.

- explain the how do you calculate the CRB in the Fig. 13 and which is the relation to the data shown in fig. 6

-add some considerations about the limit of the work.

Round 2

Reviewer 2 Report

Thank you for the improvements.